# The Medical Education Planetary Health Journey: Advancing the Agenda in the Health Professions Requires Eco-Ethical Leadership and Inclusive Collaboration

**Michelle McLean** [1,*] **, Georgia Behrens** [2] **, Hannah Chase** [3] **, Omnia El Omrani** [4] **, Finola Hackett** [5] **, Karly Hampshire** [6] **, Nuzhat Islam** [7] **, Sarah Hsu** [8] **and Natasha Sood** [9]

1   Faculty of Health Sciences & Medicine, Bond University, Gold Coast, QLD 4226, Australia
2   MSc Public Health Candidate, London School of Hygiene & Tropical Medicine, London WC1E 7HT, UK
3   Oxford University Hospital Trust, Headley Way, Headington, Oxford OX3 9DU, UK
4   Ain Shams University Hospital, Cairo 4393002, Egypt
5   Alberta Health Services, 110 Columbia Blvd—Westview Lethbridge, Lethbridge, AB T1K 6X4, Canada
6   San Francisco School of Medicine, University of California, 513 Parnassus Avenue, San Francisco, CA 94143, USA
7   Department of Medicine, University of California San Diego Health, 200 W Arbor Dr # MC8425, San Diego, CA 92103-1911, USA
8   Department of Medicine, University of California San Francisco, 1001 Potrero Ave., San Francisco, CA 94110, USA
9   Pennsylvania State College of Medicine, 700 HMC Cres Rd, Hershey, PA 17033, USA
*   Correspondence: mimclean@bond.edu.au

**Abstract:** Climate change and the declining state of the planet's ecosystems, due mainly to a global resource-driven economy and the consumptive lifestyles of the wealthy, are impacting the health and well-being of all Earth's inhabitants. Although 'planetary health' was coined in 1980, it was only in the early 2000s that a call came for a paradigm shift in medical education to include the impact of ecosystem destabilization and the increasing prevalence of vector-borne diseases. The medical education response was, however, slow, with the sustainable healthcare and climate change educational agenda driven by passionate academics and clinicians. In response, from about 2016, medical students have taken action, developing much-needed learning outcomes, resources, policies, frameworks, and an institutional audit tool. While the initial medical education focus was climate change and sustainable healthcare, more recently, with wider collaboration and engagement (Indigenous voices, students, other health professions, community), there is now planetary health momentum. This chronological account of the evolution of planetary health in medical education draws on the extant literature and our (an academic, students, and recent graduates) personal experiences and interactions. Advancing this urgent educational agenda, however, requires universities to support inclusive transdisciplinary collaboration among academics, students and communities, many of whom are already champions and eco-ethical leaders, to ensure a just and sustainable future for all of Earth's inhabitants.

**Keywords:** biodiversity loss; climate change; eco-ethical leadership; health professions education; medical education; planetary health; sustainable healthcare

## 1. Introduction: Self-Inflicted Ecological and Health Challenges

A healthy planet with stable ecosystems, clean air and water, and comfortable temperatures is necessary for the health and well-being of all of Earth's inhabitants. Human development (albeit unequal across the world) has been at the expense of our natural environment [1–5]. Wild animal populations have declined by 68% since 1970 [6], while human destruction of natural habitats has increased the frequency of spillover of zoonotic species from animals to humans, with COVID-19 being the most recent example [7,8]. With what David and colleagues have called a 'capitalist-productivist Western model' [9], a resource-driven global economy and consumptive lifestyles of the world's wealthiest

have resulted in the overshoot of several planetary boundaries, including climate change, novel entities, land change patterns, and nitrogen and phosphorus overloading of soils through agriculture [1,2,4,5]. Biophysical boundaries are being transgressed faster than they are achieving social indicators [5]. It is therefore not surprising that just prior to the 2021 Conference of Parties (COP26) in Glasgow, the International Panel on Climate Change (IPCC) and the 2021 *Lancet* Countdown on Climate Change and Health identified a Code Red for humanity and a healthy future [10–12]. The World Health Organization's (WHO) Special COP26 Report, *The Health Argument for Climate Action*, outlined 10 recommendations for urgent action on climate change, with Recommendation 7 calling for the protection and restoration of nature, the foundation of health and well-being for all of Earth's inhabitants [13]. Late in 2021, the *Geneva Charter for Well-being* highlighted the need for global commitments to achieve equitable health and social outcomes now and for future generations, without destroying the health of our planet, with *Value, respect, and nurture Earth and its ecosystems* the first of five areas for creating well-being societies [14]. This was followed in July 2022, with the United Nations General Assembly declaring access to a clean, healthy, and sustainable environment a basic human right [15]. We know, however, that in low- and middle-income countries particularly, millions of people do not have access to sufficient water (let alone clean water) or live in some of the most polluted cities in the world. As an example, a 2021 United Nations International Children's Emergency Fund (UNICEF) Agency Report identified that 2.2 billion children are disproportionately affected by changes in their environment in terms of air pollution (2 million), water scarcity (920 million), heatwaves (820 million), vector-borne diseases (600 million) and flooding (570 million) [16].

Drawing on McKimm and McLean's (an author) eco-ethical leadership [17], this viewpoint is a descriptive chronological account of the evolution of planetary health in the medical curriculum (and beyond) through the lens of a medical educator, medical students, and recent medical graduates, all of whom have been extensively involved in climate education, education for sustainable healthcare, and/or planetary health education (Supplementary material). We acknowledge the pioneering work of educators and clinicians to advance an urgent educational agenda, as well as that of medical students, who, frustrated by the slow medical education response, have, often with faculty guidance, developed guidelines, policies, and curricula to ensure that they are adequately prepared to tackle the challenges we already face in terms of a changing climate and biodiversity loss [18–20]. While we have focused on the evolution of planetary health in medical education, we acknowledge that as healthcare delivery involves different health professionals, collaboration across disciplines and generations is essential [21] if we are to address the Code Red that has been declared for humans and the planet [10–12]. The current asymmetry in healthcare needs to be balanced, and conversations democratized, with all voices contributing equally [21–26]. Most important are the voices of the traditional custodians (i.e., Indigenous Peoples of the world) of the lands on which many of us now live and work, many of whom have been dispossessed of their lands and marginalized through colonization [22–26].

## 2. Prioritizing a Key Determinant of Health and Well-Being: The Health of Our Planet (i.e., Flourishing Ecosystems and a Stable Climate)

Although Indigenous ways of knowing, doing, and being have for thousands of years been based on ontological principles of 'right relations' characterized by relationality and reciprocity amongst human and non-human relatives in our planetary ecosystems, a range of factors has disrupted and, in some instances, severed our social, spiritual and cultural connection to 'land' or 'Country' [22–26]. While colonization dispossessed many Indigenous People of their land and cultures, capitalism has commodified Earth's natural resources, which has led to ecosystem disruption, biodiversity loss, a changing climate, and air, soil, and water pollution [9,26].

Although 'planetary health' was coined in 1980 when the Friends of the Earth expanded the WHO's person-centric definition of health to include 'ecological well-being, and that personal health involves planetary health' [27], and Hancock in 1997 wrote about the need to "talk about planetary health as the ultimate determinant" of human development" [28], planetary health did not gain traction in medicine or medical education until recently. While some medical schools offered environmental health education, e.g., Williams [29], this was the exception rather than the norm [30]. In 2003, Rapport and colleagues called for a new paradigm in medical education thinking, with collaboration needed beyond the confines of the medical field because of the impact of the destabilization of natural ecosystems and the biosphere, and the increasing prevalence of vector-borne diseases on health [31,32]. Around that time, other integrative concepts and practices in transdisciplinary health, such as EcoHealth and One Health, were also being advocated, with EcoHealth included in the University of Hawaii medical curriculum [33,34]. Recent zoonotic disease outbreaks and pandemics such as Severe Acute Respiratory Syndrome (SARS), avian influenza, the 2009 H1N1 influenza, Middle East Respiratory Syndrome (MERS), Ebola, and more recently, COVID-19, have led to renewed interest in One Health in medical education [35–37].

### 3. Climate Change and Sustainable Healthcare Education Takes Center Stage (2009–2012)

From about 2007, a series of publications and global events, including the 2007 Intergovernmental Panel on Climate Change Report [38], the 2008 World Health Day focusing on climate change [39], the promulgation of the United Kingdom's 2008 Climate Change Act [40], Costello and colleagues' 2009 opening statement of the final report of a year-long Commission of *The Lancet* and University College London (UCL) Institute for Global Health that "Climate change is the biggest global health threat of the 21st century" (p. 28) [41], and the World Medical Association's 2009 Declaration of Delhi on Health and Climate Change [42], probably influenced the calls for climate change to be included in the medical curriculum. In 2009, Green and colleagues, recognizing the vulnerability of Australia to a changing climate, identified learning outcomes and potential future roles of Australian medical students in terms of climate adaptation, mitigation, resilience, advocacy, community understanding, and research [43]. In 2010, Bell, published the competencies needed to meet emerging public health issues such as climate change, writing about the need for eco-literacy in medical training (undergraduate and post-graduate), extending the disciplines of occupational and environmental medicine, the latter of which included a growing subdiscipline of 'ecosystem health' [44].

The same year that the UK's 2008 Climate Change Act was promulgated [38], the National Health Service (NHS) established a Sustainable Development Unit to provide system leadership and convene and catalyze action to reduce carbon emissions in healthcare [45]. A carbon footprint was calculated, leading to an NHS Carbon Reduction Strategy for England. The Campaign for Greener Healthcare was founded, which later became the Centre for Sustainable Healthcare (CSH), with education as a key focus [46]. The next logical step was translating this into action, primarily in medical education. In 2009, the Sustainable Healthcare Education Network was established, identifying three primary learning outcomes for health education that underpinned the CSH's educational work [47,48], culminating in 2018 with the first medical education accreditation body including 'sustainability' in a graduate outcome statement [49].

It was also in 2009 that Rockström and colleagues introduced the concept of planetary boundaries and a safe operating space for humanity [1]. The same year, the term 'The Anthropocene' was coined to describe the impact of human activities on the natural world [50], which David and colleagues remind us has been unequal, suggesting 'Capitalocene' was more appropriate [9].

For the next few years, from 2009 onwards, climate change and sustainable healthcare dominated the medical education landscape in the USA and the UK, but from about 2013, the conversation began to change, with a 'planetary health' re-awakening.

## 4. A 'Planetary Health' Re-Awakening, but Climate Change Still the Focus (2013–2017)

A White Paper emerging from the 2013 Beijing Global Health Summit highlighted that future well-being lies in the planetary health concept and associated policies. That White Paper was Horton and colleagues' 2014 'From Public Health to Planetary Health Manifesto', with the vision for "a planet that nourishes and sustains the diversity of life with which we coexist and on which we depend . . . . Planetary health is an attitude towards life and a philosophy for living . . . emphasises people, not diseases, and equity, not the creation of unjust societies . . . " [51] (p. 847). A year later (2015), The Rockefeller Foundation–*Lancet* Commission on Planetary Health was launched, with planetary health defined as: "The achievement of the highest attainable standard of health, wellbeing, and equity worldwide through judicious attention to the human systems—political, economic, and social—that shape the future of humanity and the Earth's natural systems that define the safe environmental limits within which humanity can flourish. Put simply, planetary health is the health of human civilization and the state of the natural systems on which it depends" [3] (p. 1978). The Planetary Health Alliance (PHA) was founded in 2016 on a recommendation of the Rockefeller-*Lancet* Commission on Planetary Health [52], and in 2017, the *Lancet Planetary Health* journal was launched.

Amid this planetary health re-awakening, two important 2016 publications raised awareness of the need for ecosystem knowledge and environmentally accountable medical education. Walpole and colleagues' Best Evidence Medical Education (BEME) systematic review identified what tomorrow's doctors need to learn about ecosystems and advocated for sustainable healthcare to be underpinned by the interconnection of health and the environment [53], while in an Association for Medical Education in Europe (AMEE) Guide on socially accountable medical school, Boelen and co-authors wrote that "medical schools should in future be environmentally accountable if they are to be socially accountable— the two are inextricably interlinked" (p. 1087) and offered environmental accountability criteria [54]. To this end, an environmentally accountable medical school needs clear, effective policies to reduce the school's environmental footprint and, through education, research and service, ensures health systems are sustainable for future generations. An evidence-based action plan, developed in conjunction with local communities and with measurable outcomes, should also be provided [54].

During this planetary health reawakening, a 4 December 2015 White House press release announced that the Obama Administration had signed up an additional 48 public health, medical and nursing schools from 15 countries to train their students to address the health impacts of climate change (Health Educators Climate Commitment Pledge), taking the committed schools to 118 in 29 countries [55]. Elsewhere, academic climate champions continued to advance the climate and sustainable healthcare education agenda, e.g., in 2015, Walpole and colleagues in the UK published environmentally sustainable learning outcomes following extensive consultation [48], and in Australia, Maxwell and Blashki argued for climate change and eco-literacy in the medical education [56].

## 5. Slow Uptake by Medical Schools, Leading to Student Action (2015–2017)

Despite these developments, the medical education uptake was too slow for some students. From about 2016, concerned students began taking action. The International Federation of Medical Students' Associations (IFMSA) released a 2016 Policy Statement on Climate Change and Health (updated in 2020), calling for climate change to be integrated into medical curricula [57], and developed a climate and health training manual in collaboration with the World Health Organization to encourage students to advocate for climate action. The same year, the Canadian Federation of Medical Students (CFMS) formed the Health and Environment Adaptive Response Task Force (HEART) to develop planetary health competencies for all Canadian medical schools [58,59]. The Canadian competencies not only identified Indigenous communities' vulnerabilities due to historical disenfranchisement but also included a strengths-based approach by honoring Indigenous cultural practices and ways of being. The role of Indigenous connection to the land and

related expertise was acknowledged in terms of protecting the health of humans and our natural environment.

In line with its 2017 Climate Change and Health Policy Document (updated in 2020) [60], the Australian Medical Students' Association (AMSA) launched an online peer-to-peer short course on climate change and health and ran peer-to-peer education initiatives on climate change and health, while advocating for the inclusion of climate change and planetary health in the medical curriculum. A guide to sustainable event planning followed in 2018 [61].

## 6. Medical Education Action, with Planetary Health Beginning to Gain Some Traction (2017–2018)

In the UK, medical education champions continued to drive the sustainable healthcare agenda, e.g., in 2017, Walpole and collaborators published an international collaboration to develop an environmentally accountable medical curriculum [62], and Walpole and Mortimer evaluated a collaborative project to develop sustainable healthcare education in eight UK medical schools [63]. Following a 2012 UK General Medical Council (GMC) invitation for the Sustainable Health Education (SHE) Network (coordinated by Frances Mortimer and Stefi Barna) to develop recommendations for sustainable healthcare in medical education, the SHE Network-coordinated national consultation, which included medical students, led to the GMC publishing the 2018 Graduate Outcome Statement (#25): "Newly qualified doctors must be able to apply the principles, methods, and knowledge of population health and the improvement of health and sustainable healthcare to medical practice" [49].

In the US, educators such as Arianne Teherani [64], Caroline Wellbery [65], and their colleagues began integrating sustainable healthcare and climate change into the medical curriculum. In Australia, Madden and colleagues provided recommendations for climate change and sustainability inclusions in the Australian Medical Council Accreditation Standards for Undergraduate Medical Education [66]. Recognizing that human-driven global climate change, overuse of finite natural resources, decreased biodiversity, over-population and demographic trends such as inequitable wealth distribution have brought about a crisis for environmental sustainability, the World Federation of Occupational Therapists published *Sustainability Matters: Guiding Principles for Sustainability in Occupational Therapy Practice, Education and Scholarship* in 2018 [67].

2018 saw some planetary health action in medicine. Tony Capon, the first Professor of Planetary Health (University of Sydney, Australia), and his colleagues' editorial explained what planetary health was and what doctors need to do professionally and personally [68]. A month later, Xie et al. offered a planetary health framework for primary care providers which included surveillance and reporting, patient education, health promotion and advocacy in terms of climate action [69].

## 7. Advocating for Strengths-Based 'Planetary Health' Incorporating Traditional (Indigenous) Ways of Knowing, Doing, and Being (2018–2020)

From about 2018, Indigenous voices were finally being heard in the 'planetary health' space [22–25,70–72]. In a 2018 Special Issue of Challenges (The Emerging Concept of Planetary Health: Connecting People, Place, Purpose and Planet), Prescott and colleagues wrote "Planetary health is not a new discipline; it is an extension of a concept understood by our ancestors and remains the vocation of multiple disciplines" [70] (p. 1). Additionally, published was the *Canmore Declaration: Statement of Principles for Planetary Health*, a consensus statement that expanded the Ottawa Charter for Health Promotion, which affirmed the need to consider the health of people, places, and the planet as indistinguishable [23]. That 'planetary health' was not a new 'discipline' was recently reiterated by Nicole Redvers, a First Nation Canadian educator, and clinician: "Planetary health as a "field" is primarily a Western construct as Indigenous Traditional Knowledge systems have no clear separation of self or that of the community and the ecosystem at large (5). This means that the meaning and

application of planetary health are directly rooted in community values based on protocols for living in harmony with all that have existed for thousands of years (6)" [72] (p. 2).

The planetary health education momentum was finally taking off, not just in medical education but across all health professions education. In May 2018, Stone and colleagues published 12 cross-cutting principles for planetary health education to address the urgency, skills required, global citizenship, inequity, and inequality [73]. Principle 12 (Historical and current global values) related to marginalized voices: "An understanding of the past is necessary to solve the problems of the present. To grasp the necessity and urgency of planetary health, students need to be aware of the historical perspectives and milestones that have laid the foundation for the field, including those perspectives that have been historically marginalized or ignored. To identify opportunities for positive interventions, students must recognize patterns over time and appreciate current global context" (p. 193).

Nursing and nursing education have been particularly active in the planetary health space. Following a 2017 call to action for planetary health nursing [74], Rosa and Upvall called for a paradigm shift from global nursing to planetary nursing in 2019 [75], and Potter advocated for planetary health nursing education [76]. Around the same time (2019), a new field of physiotherapy emerged, Environmental Physiology (EPT), in which the key aspects of the relationship between the environment, human health, and functioning and physiotherapy are considered and respected to mutually benefit patient health, the physiotherapist and the environment. EPT covers areas of clinical practice, research, and education, with the need to learn from traditional and Indigenous ways of life, knowledge, and how they have integrated environment, health, and community with each other [77,78].

In a 2020 *Medical Teacher* Special Issue, Redvers and her fellow Indigenous scholars' seminal article, which won an Association for the Advancement of Sustainability in Higher Education award in December 2021 [79], provided Indigenous perspectives on education for sustainable healthcare [71]. The article describes how education for sustainable healthcare is embedded within planetary health and Traditional Knowledge and Natural Laws. The authors offer eight recommendations for the Indigenization of sustainable healthcare and planetary health education such as prioritizing the intake of Indigenous students and Indigenous staff appointments, transforming current sustainability frameworks to include an Indigenous-led planetary health lens, and centralizing Indigenous-led 'land' and 'Country'-based understanding of health and well-being in curricula.

### 8. Two Significant Years for Climate Change and Planetary Health Education (2019–2020)

2019 and 2020 were significant years for climate change and planetary health education, the agendas being driven to a large extent by medical students. Key activities in climate change education include:

- 2019: Building on the 2015 pledge by 115 medical, nursing, and public health schools in North America, South America, Asia, Europe, Africa, and Australia to add climate and health to their curricula [55], with Rockefeller Foundation funding, Columbia University's Mailman School of Public Health, home to the nation's first academic program in climate and health, announced the launch of a Global Consortium on Climate and Health Education (GCCHE) to share best scientific and educational practices and design model curricula on the health impacts of climate change [80].
- 2019: The American Medical Association passed a resolution for the inclusion of climate change across the education continuum [81].
- 2019: Clinicians for Planetary Health was launched, with a call for collective action for all clinicians (doctors, nurses, etc.) to consider their dietary choices, modes of transport, energy sources, political action, community engagement, and environmental stewardship [82].
- 2020: Medical students continued to advocate for climate change education [83], including AMSA, which published an updated Climate Change and Health Policy, calling for Australian universities to integrate climate change and its relationship to health in the medical curriculum and other health courses [84]. Importantly, that policy

identified the need to work in consultation with Aboriginal and Torres Strait Islander communities to ensure their concerns were heard and acted upon, recognizing that their identity is intrinsically linked to Country, with a need to respect and abide by self-determination and cultural lore.

Perhaps spurred by COVID-19, with its links to our poor relationship with our natural world [85], the deteriorating state of our environment, and several WHO and UN publications and campaigns, such as the 2020 *Manifesto for a Healthy Recovery from COVID-19* [86], and the 2021–2030 *Decade of Environment Restoration* [87], planetary health inclusion in medical education progressed during 2019 and 2020:

- 2019: Walpole and other health professionals described planetary health integration in clinical education [88]
- 2019–2020: With a desire to hasten curriculum inclusion of planetary health, more sustainable and inclusive research and campus practices, community outreach, and support for student initiatives, the student-led Planetary Health Report Card (PHRC) was launched [89,90], with two authors being co-founders (Karly Hampshire and Nuzhat Islam). Health professions education can use the PHRC as a needs assessment to promote institutional change. The first audit cycle (2020) included North American medical schools [89,90].
- 2019: Medical Students for a Sustainable Future (MS4SF) was founded in the US by two authors (Natasha Sood and Sarah Hsu). MS4SF aims to unite medical students invested in the health of our planet and patients, developing resources, celebrating Earth Day and supporting the PHRC [91,92].
- 2019: Prescott and colleagues wrote the following: "We argue that in 2019, one simply cannot claim to be a 'health' care professional without advocating forcefully for the planet" [93].
- 2020: Inspired by climate change, sustainable healthcare, and planetary health education champions, which included student advocates and activists, McKimm and McLean published their eco-ethical leadership article to drive this urgent agenda to ensure a just and sustainable future [17].
- 2020: Following IFMSA's 2019–2020 survey of 2817 medical schools in 112 countries which found that only 15% of medical schools included climate change and even fewer (11%) included air pollution in the curriculum [94], medical students called for the integration of planetary health in every medical curriculum [94,95], as did McLean and colleagues who co-edited a 2020 special *Medical Teacher* issue [96].
- 2020: Parkes and colleagues wrote that in preparing for an eco-social approach to public health education, "Health professionals will be called on to engage as humble, informed, and trusted partners in the collective, boundary-crossing effort of transforming practices and structures to better sustain the health and well-being of all life, including our own" [97] (p. 4).
- 2020: A Planetary Health Pledge was published with the aim to unite all health professionals and students around the interconnection of health and well-being at a personal and planetary level [19].

### 9. Planetary Health: Is It Becoming Mainstream in Health Professions Education?

The planetary health momentum continued, with 2021 a key 'planetary health' year not only for medical education but for all health professions education and higher education in general. In February 2021, the United Nations Environment Programme published *Making Peace With Nature*: *A scientific blueprint to tackle the climate, biodiversity and pollution emergencies* [98], a conversation that would undoubtedly have influenced planetary health education such as the three key education publications, with Indigenous authors and an acknowledgment of the need to include Traditional Knowledges and practices:

- March 2021: The AMEE Consensus Statement on Planetary Health and Education of Sustainable Healthcare, with 35 collaborators from various health professions, including students representing different regions [99].

- April 2021: On Earth Day (22 April), the Planetary Health Alliance launched the Planetary Health Education Framework, which reflected the collaborative input of an interdisciplinary, intergenerational, geographically diverse, and gender-inclusive task force from the field of planetary health, including education [100].
- October 2021: The 2021 São Paulo Declaration on Planetary Health called for planetary health citizenship across higher education [101].

Another PHRC cycle was completed in 2021. Under the leadership of Hannah Chase (an author), 90% of UK and the Republic of Ireland medical schools undertook the self-audit [102]. With the need to apply a strengths-based approach to planetary health by incorporating Indigenous ways of knowing, doing, and being, the 2021 cycle included the following Curriculum metric: *To what extent does your medical school emphasize the importance of Indigenous knowledge and value systems as essential components of planetary health solutions?* The PHRC continues to expand internationally. Medical school report cards are available for Germany, Japan, India, and Malaysia, with 2022 self-audits underway in South Africa, New Zealand, Portugal, Switzerland, Greece, Turkey, Japan, and Indonesia. Recognizing that planetary health solutions require engagement from all health professionals, there are now PHRC metrics for pharmacy and nursing, with metrics for physiotherapy and dentistry in development. The PHRC has inspired increased engagement in planetary health and sustainable healthcare at many institutions, including core curricula transformation, assessment, creation of student-faculty task forces, new community partnerships, and new sustainability websites [89,90,102].

Organizations such as inVIVO [103] and the Planetary Health Alliance [52], in particular, the regional hubs, have facilitated the inclusion of all voices, particularly those from the 'Global South', which have generally been neglected. In the Philippines, for example, Renzo Guinto has been leading a robust planetary health education focus, with the Planetary and Global Health Program at St. Luke's College of Medicine [104] and the Applied Microbiology for Health and Environment Research Group at the University of the Philippines, Manila [105]. In Malaysia, a Special Interest Group of IFMSA Asia-Pacific ran a four-week Planetary Health course in 2021 [106].

Just prior to the COP26 in November 2021, the same editorial—*Call for emergency action to limit global temperature increases, restore biodiversity and protect health*—appeared in numerous health journals [107]. The editors wrote: "Ahead of these pivotal meetings, we—the editors of health journals worldwide—call for urgent action to keep average global temperature increases below 1.5 °C, halt the destruction of nature and protect health . . . . As health professionals, we must do all we can to aid the transition to a sustainable, fairer, resilient and healthier world. Alongside acting to reduce the harm from the environmental crisis, we should proactively contribute to global prevention of further damage and action on the root causes of the crisis".

Based on the deteriorating state of our natural environment and COVID-19 (which has not gone away), there were and still are loud calls to include planetary health in the medical (and other health professions) curriculum, which are continuing in 2022 [12,21,74–76,108–112], including Wabnitz and Guzmán advocating for liberal education to prepare planetary stewards [113]. Recent published examples of planetary health integration in medical education include Capetola, Noy, and Patrick's planetary pedagogy [114], Sliming and colleagues' advocacy curriculum [115], a student-driven planetary health curriculum [116], and the outcomes of a five-year planetary health assignment using the Sustainable Development Goals, SDG 13 (Climate action) in particular [117].

Progress is also being made in post-graduate medicine and continuing professional development. To this end, Cooke and colleagues have reported sustainability outcomes for some UK medical colleges [118], while Shah has described how the values such as inclusivity relating to planetary health education described in the AMEE Consensus Statement have been integrated into the UK's Royal College of General Practitioners curriculum to address systemic injustices such as the marginalization of certain knowledges, learning, world views or health beliefs [119]. Asaduzzamen and co-authors have published an extensive

list of planetary health post-graduate and continuing professional courses, but noted that most were offered in the Global North and mainly aimed at doctors [120].

## 10. Progressing Planetary Health in Health Professions Education

Time is running out, with tipping points fast approaching [121]. In addition, Mora and colleagues warn that over half of the known pathogenic diseases can be exacerbated by climate change [122]. We thus need to collaborate to find a safe and just corridor for people and the planet [123], drawing on all intergenerational sources (including Indigenous) of insight, experience, knowledge, wisdom, expertise, passion, and a genuine concern for a sustainable future not just for humans but for all 'beings' [22–25,71,72]. To ensure that all health professional students are prepared for the current ecological crisis and to inculcate the appropriate values, universities and health professions institutions need to be socially and environmentally accountable [54]. Mission and vision statements should reflect and provide evidence of how historical and current inequities (environmental and social determinants of health, racism, gender, etc.) are being addressed in admission policies, appointments, learning and teaching, research, community engagement, and general operations, as well as having a roadmap to meet emissions reductions targets. While, for example, the AMEE ASPIRE Excellence Social Accountability Award includes a criterion for environmental accountability [124], this should be a requirement for health professions education rather 'nice to have'. The global uptake of health professions education institutions to participate in the student-led PHRC demonstrates a willingness to engage in collaborative self-improvement [89,90,102]. Institutions need to 'walk the talk' by not only educating for planetary stewardship [113] but also ensuring ecosystem restoration through inclusive consultation and collaboration [99,107,113].

Institutions need to foster faculty and student collaboration in curriculum renewal [125,126]. Students bring fresh ideas and, in many instances, have deeper knowledge of the climate and ecological crisis, while academics have a better understanding of the education system. Tun and colleagues described how partnerships between medical students and faculty can enhance faculty development in planetary health and sustainable healthcare [126], while Navarrete-Welton and fellow students were supported in developing their planetary health curriculum by academic faculty [116].

If not already members, students and faculty can join groups such as the IFMSA and Clinicians for Planetary Health to advance allyship in terms of ecological justice and systemic change to address historic marginalization and injustice. This collaboration can become more universal by taking the 2020 Planetary Health Pledge [19], jointly accepting responsibility and taking the lead to protect the health of communities and the planet to ensure just, equitable and access to natural resources, embracing all forms of 'knowing' doing and being, including Indigenous Traditional Knowledges [17].

Berwick, in his 2020 essay on the moral determinants of health made a passionate plea for health professionals to become involved in campaigning on moral issues such as human rights, racial injustice and climate change, writing "When the fabric of communities on which health depends is torn, then healers [health professionals] are called to mend it. The moral law within insists so" [127] (p. 226). Launer predicted that as health and social inequalities widen, "doctors around the world will be drawn inescapably into political campaigning" [128] (p. 611). Recent examples of health activism include the 46 million health workers who directed their call for global climate action to COP26 and COP27 Presidencies [129] and the 2022 global memorandum from more than 1000 health professionals and 200 health organizations calling on governments to urgently develop and implement a Fossil Fuel Non-Proliferation Treaty to end global dependence on fossil fuels, in order to protect global public health [130]. Clery and colleagues have called on medical training and regulatory bodies to support health activism as health professionals have a moral obligation to act for the wider determinants of health on issues that cross socio-politico-environmental boundaries, particularly in terms of the most vulnerable in

society [131], while Krzanowski, recognizing the entwined crises of climate change and ecological collapse, has called for biodiversity champions in psychiatry [132].

At COP26, the UN Secretary General, António Guterres, said that "The climate action army—led by young people—is unstoppable. They are larger. They are louder. Additionally, I assure you, they are not going away. I stand with them" [133]. As agents of change [133,134], not just medical students but all health professional students, as future health professionals, have the power to advance their own education, as we have described. They also have the power to be advocates and activists to address not only the existential threat to all of Earth's inhabitants, but particularly in countries or regions where colonization has marginalized Indigenous Peoples, students will also need to be allies, accomplices and co-conspirators [135] to ensure that traditional ways of doing, being and knowing inform clinical practice and that human rights are applied to policy decisions. For Prescott and colleagues, all these efforts must be underpinned by an intergenerational justice perspective, recognizing that tomorrow's health depends on the choices we make today [136]. This appears to be coming to pass, with young activists already prioritizing human rights in the climate change agenda [137]. The UN has recognized the need for intergenerational representation. After setting up a Youth Advisory Group on Climate Change in 2020, a UN General Assembly resolution in September 2022 established a UN Youth Office in the Secretariat dedicated to youth issues in areas such as peace and security, sustainable development and human rights [138]. In September 2023 at the 54th session of the Human Rights Council, there will be a panel on youth and human rights [139]. The voices of young people were heard at COP27 (November 2022), with the first-ever Youth Envoy being one of the authors, Omnia El Omrani. COP27 also had an independent and inclusive children and youth pavilion for the first time. A COP27 milestone was that young people became official stakeholders in climate policy under the Action for Climate Empowerment (ACE) action plan, which was outlined in Article 12 of the 2015 Paris Agreement [140]. As an example of youth engagement, during COP27, UNICEF released a WHO Policy Brief, *Climate Change, Health, & Intergenerational Equity*, which included input from the Young Doctors Health Working Group and IFMSA [141].

## 11. Concluding Comments

This viewpoint traces the chronological evolution of planetary health in medical education, which we believe should now be mainstream in all health professions education. The deteriorated state of the planet's biosphere requires that professionals be advocates (and in some cases, activists) for all of Earth's inhabitants now and in the future. In our various educational contexts, we have advocated (and in many instances taken action) for all health professional students to be prepared to deal with a changing climate as well as reduce healthcare's environmental footprint. We extend this responsibility to stewardship and restoration of the planet's ecosystems, in line with Indigenous Peoples' ways of knowing, doing, and being. With the UN General Assembly declaring access to a clean, healthy, and sustainable environment a human right in July 2022 [15], this should hopefully galvanize action with respect to advocacy and policy action [142], including in health professions education.

David and colleagues, however, remind us of the very unequal contribution to the 'Anthropocene', probably more appropriately the 'Capitalocene: " . . . it is not humanity in a generic and abstract sense that is responsible for contemporary devastation. Rather, it is the capitalist model of society centered on unlimited growth and the ever-more intensive exploration of natural resources . . . Incriminating humanity masks the responsibility of the productivist Western model and contributes more often than not to stigmatising the world's most vulnerable populations" [9] (p. 1146). At COP27 in November 2022, poorer nations succeeded in committing wealthier nations to a 'loss and damage' fund to compensate 'developing' countries for the harm to the climate caused by 'developed' nations [143]. Time will tell if this will be honored. That 90% of the 9 million deaths each year from pollution (air, lead, chemical) occur in low-income countries [144] is a stark reminder of the

environmental and social inequity and injustice that exists, which will be exacerbated by a changing climate and environmental degradation for the Most Affected People and Areas (MAPA), i.e., women, Indigenous Peoples, racialized people, LBTQ+ people, the young, those facing social deprivation and the Global South [145].

Along our individual journeys (Supplementary material), we have been inspired and mentored by clinicians, educators, peers, colleagues, and community members who are eco-ethical leaders working towards a just and sustainable future [17]. We hope that, like them, we can continue to advance this urgent agenda and inspire the next generation of health professionals to continue to be proactive. That there are learning outcomes in, for example, medical education accreditation standards or national licensing examinations relating to sustainability, such as in the UK [49] or health and the climate crisis in Canada [146], is credited to the ongoing advocacy of faculty and student champions. Much more, however, needs to be done, and urgently.

We acknowledge that this account from the extant literature and our personal reflections may have unintentionally excluded the voices of medical students and faculty champions and leaders in the 'Global South' who may not have access to the same resources (e.g., Open Access publication funding, stable Internet access) available to students, faculty, and communities in the wealthier 'Global North'. Our challenge is to ensure regional and intergenerational inclusivity for a just and sustainable future for all.

**Supplementary Materials:** The following supporting information can be downloaded at: https://www.mdpi.com/article/10.3390/challe13020062/s1, Supplementary material: Author autobiographies (Supplement S1) and Student Organizations (Supplement S2).

**Author Contributions:** M.M., G.B., H.C., O.E.O., F.H., K.H., N.I., S.H. and N.S. contributed to the writing and editing of this manuscript over several months. All authors have read and agreed to the published version of the manuscript.

**Funding:** This research received no external funding.

**Data Availability Statement:** Not applicable.

**Acknowledgments:** We would like to acknowledge the many individuals who have inspired us to become involved in advancing this urgent agenda. They include faculty members, colleagues, and peers who have mentored, coached, encouraged, and collaborated with us.

**Conflicts of Interest:** The authors declare no conflict of interest.

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
