# Peer review of "The Medical Education Planetary Health Journey: Advancing the Agenda in the Health Professions Requires Eco-Ethical Leadership and Inclusive Collaboration"

_challenges, doi:10.3390/challe13020062_

Round 1

Reviewer 1 Report

McLean and colleagues are to be commended for synthesizing a vast literature and providing what appears to be the first concise historical timeline of ‘how we got to here’ re: planetary health. The manuscript is well-written, the appropriate background is provided (to underscore the urgency), and builds up section by section in a logical manner. Readers from all branches of medicine and allied health professionals, especially educators, should see the relevance and urgency of the discussion. This is a unique blend of authors (demonstrated by the Supplementary bios) who have found common ground in their own diversity, and together they have made a cogent argument for the mainstreaming of planetary health in medical education.

The manuscript is a good fit for Challenges and is near-ready for publication. Here are some notes for the authors. 

This manuscript might be better positioned (i.e., labeled through the submission system) as a ‘Viewpoint’ rather than a ‘Review’ paper. The authors concede up front, in the Abstract, that their narrative review of the historical literature is mixed with personal experience and personal interactions. This concession isn’t a bad thing. On the contrary, it is the strength of the article – at the same time, though, especially with the tone (as a product of those experiences) it shifts the manuscript position from a classic ‘Review’ paper.

Abstract Lines 30-31 -  “developed learning outcomes, resources, policies, frameworks, and institutional audit tools”. Please qualify this sentence. Pertaining to what? Planetary health?

Line 66 – “this submission”. Please change to something like “this Viewpoint” or “this manuscript” or “our contention here…”

Here are some general thoughts that might tighten the manuscript and develop specificity toward what a revised planetary health education might look like:

In the opening line of the Abstract it is stated that the planet’s ecosystems are threatened by  a “resource-driven economy and the consumptive lifestyles of the wealthy”. This is a brilliant open, but was never expanded upon in the body of the manuscript, save for a repeat of this line (lines 48-50). I would suggest that the authors expand on this reality in relation to what the specifics of planetary health medical education might look like. Where does neoliberalism enter the discussion? Are elite private universities contributing to inequity and marginalization by virtue of their very own elite status? What about the commercial determinants of health? The global purveyors of unhealthy product, including ultra-processed foods, are often deeply embedded in Global North medical schools (in plain sight, right in the cafeteria). How does this square with educational efforts that seek to promote health along the personal/public/planet continuum?

In the paragraph before the Conclusion, it looks like the authors are just about to broach this deeper subject, but then it gets diverted to a significant word-count being chewed up by discussion of psychiatry. While it is true that professionals in psychiatry are well placed to be part of a planetary health educational movement, they are not alone in this. Left as is, the paragraph might infer that psychiatrists should be default leaders or privileged in planetary health medical education or discourse. Every medical and allied health discipline is equally well positioned to be champions of biodiversity. I don’t think the paragraph needs to be dialed back or references to psychiatry removed – it needs to run deeper, and match the courage demonstrated in the opening line of the Abstract. It is the ideal spot to discuss the commercial determinants of health and the upstream drivers of the consumptive lifestyle. Who is pushing the global desire to be one of the wealthy consumers? Could it be the very same groups that donate handsomely to Global North universities and medical schools, yet at the same time sustain the inequalities spoken of in that paragraph?  Have a look at  David PM, Le Dévédec N, Alary A. Pandemics in the age of the Anthropocene: Is 'planetary health' the answer? Glob Public Health. 2021 Aug-Sep;16(8-9):1141-1154 for a critical discussion https://pubmed.ncbi.nlm.nih.gov/33635169/

In order to strengthen the link between planetary health medical education and the daily work of healthcare providers, it might be worth considering a brief discussion of some of these references –

 https://www.thelancet.com/journals/lanplh/article/PIIS2542-5196(18)30055-X/fulltext

https://www.thelancet.com/journals/lancet/article/PIIS0140-6736(19)30846-3/fulltext?source=post_page---------------------------

Author Response

Dear Reviewer 1, I have attached a Response to Reviewers file. Thank you for the excellent feedback. We believe that we have addressed your comments. 

Reviewer 2 Report

This is a well-made and factful piece about the emergence of planetary health in medical education. For a reader who is not well acquainted with the concept, it is an insightful read into an important topic. A strength of the paper is that it reveals how different kinds of actors contributed to the overall process. It is commendable that the authors emphasize the important role of students. 

What is missing, however, is a regard for the social aspects of planetary health. As the authors make clear, planetary health includes equity and healthy relations between people. However, the historical account focuses on the ecological dimension of planetary health education exclusively (e.g. climate change). Social aspects are only addressed when pointing to the marginalization and important role of Indigenous People. Yet, I think, the social dimension of planetary health is much broader. For example, Marmot’s study about the status syndrome shows that people with inferior organizational status have poorer health outcome in comparison to their superiors. Therefore, the authors should add information on the ways in which the social dimensions/the social bases of (planetary) health have been considered by advocates and in medical education.

Author Response

Dear Reviewer 2, attached is our Response to Reviewers. Thank you for your comments. We believe that in addressing Reviewer 1 comments, we have also addressed your comments. 
